# Exploration of Emotion Dynamics Sensing Using Trapezius EMG and Fingertip Temperature

**DOI:** 10.3390/s22176553

**Published:** 2022-08-30

**Authors:** Wataru Sato, Takanori Kochiyama

**Affiliations:** 1Psychological Process Research Team, Guardian Robot Project, RIKEN, 2-2-2 Hikaridai, Seika-cho, Soraku-gun, Kyoto 619-0288, Japan; 2Brain Activity Imaging Center, ATR-Promotions, Inc., 2-2-2 Hikaridai, Seika-cho, Soraku-gun, Kyoto 619-0288, Japan

**Keywords:** arousal, film, fingertip temperature, trapezius electromyography (EMG), valence

## Abstract

Exploration of the physiological signals associated with subjective emotional dynamics has practical significance. Previous studies have reported that the dynamics of subjective emotional valence and arousal can be assessed using facial electromyography (EMG) and electrodermal activity (EDA), respectively. However, it remains unknown whether other methods can assess emotion dynamics. To investigate this, EMG of the trapezius muscle and fingertip temperature were tested. These measures, as well as facial EMG of the corrugator supercilii and zygomatic major muscles, EDA (skin conductance level) of the palm, and continuous ratings of subjective emotional valence and arousal, were recorded while participants (*n* = 30) viewed emotional film clips. Intra-individual subjective–physiological associations were assessed using correlation analysis and linear and polynomial regression models. Valence ratings were linearly associated with corrugator and zygomatic EMG; however, trapezius EMG was not related, linearly or curvilinearly. Arousal ratings were linearly associated with EDA and fingertip temperature but were not linearly or curvilinearly related with trapezius EMG. These data suggest that fingertip temperature can be used to assess the dynamics of subjective emotional arousal.

## 1. Introduction

Emotional experiences, which are the key component of subjective happiness [1], vary dynamically in everyday life [2]. Several previous psychological studies capturing continuous ratings have shown that emotional experiences change from moment to moment during the presentation of emotional stimuli, which diverge from the overall ratings after the stimulus presentation [3,4,5,6]. The studies further showed that such short-term emotional dynamics have unique associations with long-term psychological well-being and psychopathology (e.g., there is a positive relationship between large moment-to-moment emotional fluctuations and maladaptive psychological functioning) [7].

Because continuous recording of subjective emotional experiences is difficult in some situations (e.g., while conducting other tasks), physiological signals associated with emotional experiences can be useful to estimate subjective emotional dynamics [8]. A previous study has revealed that physiological signals, including facial electromyography (EMG) of the corrugator supercilii and zygomatic major muscles, electrodermal activity (EDA), and nose-tip temperature can be used to assess temporal changes or dynamics of subjective emotional valence and arousal [9]. Corrugator and zygomatic EMG activities reflect the emotion-related facial actions of brow-lowering and lip corner-pulling muscle activity, respectively [10]. EDA activation and reduced peripheral skin temperature reflect activity in the sympathetic branch of the autonomic nervous system [11,12]. In the study by Sato and colleagues [9], continuous ratings of valence and arousal and these physiological signals were assessed while participants viewed emotional films. The results showed that the corrugator and zygomatic EMG were negatively and positively associated with continuous valence ratings, respectively; the EDA and nose-tip temperature were positively and negatively associated with continuous arousal ratings, respectively. Those results are consistent with the findings of another study reporting intra-individual associations between continuous valence ratings and facial EMG activity during watching emotional films and playing a game involving bodily movements [13]. Other studies also suggested an association between continuous valence ratings and facial EMG signals, although the studies did not assess intra-individual associations [14,15]. Collectively, these data suggest that subjective emotional valence and arousal dynamics can be assessed based on these physiological measures.

However, it remains unknown whether other measures can be used to assess emotional dynamics. Alternative physiological signals may be more desirable under some conditions. For example, the use of electrodes on the face to record facial EMG may be visually disturbing for natural face-to-face communication.

EMG of the trapezius muscle is a candidate method for such assessment. Several previous studies found that trapezius EMG activity increased while participants watched threatening films [16] and engaged in stress-inducing tasks (e.g., mental arithmetic [17,18,19,20,21]; for a review, see [22]). Although none of these studies investigated associations between trapezius EMG activity and subjective valence or arousal dynamics, one study reported that erotic films, which induced sexual arousal, did not obviously increase trapezius EMG activity [16]. These data suggest that trapezius EMG could be used to assess emotional valence. At the same time, some studies [17,19] reported that trapezius EMG activity was positively correlated with several sympathetic nervous activity parameters (e.g., systolic blood pressure) during exposure to stress, suggesting that trapezius EMG activity may reflect subjective arousal. Based on these findings, the trapezius EMG was hypothesized to be negatively associated with continuous ratings of valence, or positively associated with continuous ratings of arousal.

Fingertip temperature is another candidate parameter for the assessment of emotion dynamics. Previous studies have reported that fingertip temperature decreased while participants watched fear-inducing films [23,24,25] and recalled or imagined fearful scenes [26,27], and increased while participants smelled a sedative odor [28]. To date, no studies tested the associations between fingertip temperature and continuous emotional ratings. Together with the aforementioned evidence that peripheral skin temperature reflects activity in the sympathetic nervous system [12], which generally reflects subjective arousal [29], and the finding that temperature in a different location was associated with arousal dynamics [9], fingertip temperature was hypothesized to have a negative association with continuous arousal ratings.

A further issue with previous investigation of the physiological correlates of subjective emotional dynamics is the assumption of linear relationships [9,10]; while zero-order correlations between continuous valence/arousal ratings and facial EMG/EDA were assessed, non-linear relationships were not. Although linear analysis is an important statistical analysis method, non-linear relationships of subjective emotional dynamics are also worth exploring regarding these physiological measures as well as trapezius EMG and fingertip temperature.

To test these hypotheses, trapezius EMG and fingertip temperature, as well as corrugator and zygomatic EMG and EDA, were measured while participants viewed five emotional clips. As in a previous study [9], after participants had watched each film, they provided overall valence and arousal ratings using an affect grid [30]. After the initial viewing with physiological recordings, the clips were shown twice more; participants recalled and continuously provided valence or arousal ratings using a slider-type affect rating dial [31]. Intra-individual correlations [32,33] between the participant’s subjective ratings and physiological activity were calculated; the strength of and differences between correlations were assessed at the group level using two-stage random-effects analyses [34]. Furthermore, polynomial regression modeling [35] was performed, again using two-stage random-effects analyses.

## 2. Materials and Methods

### 2.1. Participants

This study recruited 30 Japanese volunteers (16 women; mean ± standard deviation [*SD*] age, 22.6 ± 2.7 years). The sample size was determined through *a priori* power analysis using G*Power software 3.1.9.2 [36], based on a previous study that used similar film stimuli along with recordings of facial EMG and palm EDA [9]. Analysis of subjective–physiological concordance using a two-step procedure with one-sample *t*-tests was planned. The effect size *d* of 1.1 was estimated from the results, based on the weakest subjective–physiological concordance; an *α* level of 0.05 and a power (1-*β*) of 0.95 were used. The power analysis showed that more than 13 participants were needed. All participants had normal or corrected-to-normal visual acuity. After an explanation of the experimental procedure, all participants provided informed consent. This study was approved by the Ethics Committee of RIKEN. The experiment was conducted in accordance with institutional ethical guidelines and the Declaration of Helsinki.

### 2.2. Apparatus

The apparatus, stimuli, and procedure were the same as those reported in a previous study [9]. Specifically, stimuli were presented using a Windows computer (HP Z200 SFF; Hewlett-Packard Japan, Tokyo, Japan), a 19-inch cathode ray tube display (HM903D-A; Iiyama, Tokyo, Japan), and Presentation software (Neurobehavioral Systems, Berkeley, CA, USA). An additional laptop Windows computer (CF-SV8; Panasonic, Tokyo, Japan) and wired optical mouse (MS116, Dell, Round Rock, TX, USA) were used for the cued-recall continuous ratings. A previous technical report revealed that the response time delay for the wired optical mouse was <20 ms [37].

### 2.3. Stimuli

Three films developed in a previous study [38] were used as highly negative (anger), moderately negative (sadness), and neutral stimuli; these stimuli were previously validated by a rating task for discrete [38,39] and dimensional emotions [39]. Because the moderately positive (contentment) stimulus used in the previous study [38] had low resolution, and the highly positive (amusement) stimulus lacked a Japanese-dubbed version, they were replaced with comparable stimuli from commercial films; i.e., a waterside scene with birds (Wild Birds of Japan; Synforest, Tokyo, Japan) and a comedy dialogue between two people (M-1 Grand Prix The Best 2007–2009; Yoshimoto, Tokyo, Japan), respectively. The mean ± *SD* time of film stimuli presentation was 168.4 ± 17.8 s (range: 150–196 s). To confirm that the new stimuli elicited the intended emotions, we conducted a preliminary rating experiment with 11 participants (four females; mean ± *SD* age, 24.6 ± 8.3 years), none of whom participated in the subsequent physiological activity tests. The film clips were presented in the same manner as in the main experiment. The participants were instructed to select the best label from the following 16 discrete emotions used in previous validation studies [38,39]: amusement, anger, arousal, confusion, contempt, contentment, disgust, embarrassment, fear, happiness, interest, pain, relief, sadness, surprise, and tension. The results ensured that the target emotions were the most frequently selected for both contentment (81.8%) and amusement (90.9%) films. Besides, the most frequently selected labels were anger for the anger film (36.4%), sadness for the sadness film (90.9%), and tension and embarrassment for the neutral film (each 27.3%). However, note that the labels were used only as the film names, because the clips could elicit multiple emotional responses, as reported in previous studies using ratings based on discrete emotions [38,39]; we were interested in dimensional emotional states. A fear-inducing film used in a previous study [38] was employed in a practice trial. The stimuli dimensions were 640 horizontal × 480 vertical pixels (25.5° horizontal × 11° vertical visual angle).

### 2.4. Procedure

Experiments were performed in a soundproof, electrically shielded chamber. The temperature was maintained at 23.5–24.5 °C and monitored using a TR-76Ui data logger (T&D Corp., Matsumoto, Japan). Participants were informed that the purpose of the study was to obtain subjective ratings and record electric activity from the skin while viewing films. Participants were given approximately 10 min to acclimate to the chamber. After one practice film, five test films were presented. The order of film presentation was pseudorandomized.

For each trial, after a 1-s fixation point and 10-s white screen (pre-trial baseline), each film was presented. After another 10-s white screen (post-trial baseline), the affect grid [30] was presented to allow assessment of emotional valence and arousal using a nine-point scale. Participants were instructed to attend to the fixation point, watch the film, and rate their overall subjective experience (valence and arousal) while watching the film by pressing keys. After the responses, the screen went black during the interval preceding the next trial (randomly set to 24–30 s). Physiological data were continuously recorded during all trials.

After all trials had been completed, all stimuli were presented on the monitor twice more, while the nine-point scales for valence or arousal were simultaneously displayed on another laptop. Participants were instructed to recall their subjective emotional experience during the initial viewing, and continuously rate that experience in terms of the valence or arousal dimensions by moving the mouse. The coordinates of the mouse were sampled at 10 Hz. The participants rated valence first, and then arousal. This cued-recall procedure was used to acquire continuous ratings of valence and arousal, which were difficult to simultaneously assess during the initial viewing. Previous studies reported that cued-recall continuous ratings were strongly positively correlated with on-line continuous ratings for emotional films [9,40].

### 2.5. Physiological Data Recording

EMG data were recorded from the corrugator supercilii, zygomatic major, and trapezius muscles on the left side. Pairs of pre-gelled, self-adhesive 0.7 cm Ag/AgCl electrodes with 1.5-cm inter-electrode spacing (Prokidai, Tsuzuki, Japan) were used. The electrodes were placed in accordance with previously established guidelines [41,42,43]. Specifically, the center of the electrodes was set 2 cm lateral to the midpoint between the acromion and spinous processes of the seventh cervical vertebra for trapezius EMG recording [43]. A ground electrode was placed on the middle of the forehead. The data were amplified, filtered online (band pass: 20–400 Hz), and sampled at 1000 Hz using an EMG-025 amplifier (Harada Electronic Industry, Sapporo, Japan), the PowerLab 16/35 data acquisition system and LabChart Pro 8.0 software (ADInstruments, Dunedin, New Zealand). A low-cut filter (20 Hz) was applied to remove motion artifacts [44,45]. Video was recorded unobtrusively using a digital web camera (HD1080P; Logicool, Tokyo, Japan) to check for motion artifacts.

EDA was recorded from the palmar surface of the medial phalanges of the index and middle fingers of each participant’s left hand using pre-gelled, self-adhesive 1.0 cm Ag/AgCl electrodes (Vitrode F; Nihon Koden, Tokyo, Japan) in accordance with previously published guidelines [11]. Skin conductance level was measured by applying a constant voltage of 0.5 V using a Model 2701 BioDerm Skin Conductance Meter (UFI, Morro Bay, CA, USA). Finger temperature data were recorded from the palmar surface of the distal phalanges of the fifth finger of each participant’s left hand using an ML309 Thermistor Pod (ADInstruments). The EDA and finger temperature data were recorded using the same data acquisition system and recording software applied for the aforementioned EMG, except that no online filter was used.

### 2.6. Data Analysis

*Preprocessing.* Preprocessing was conducted in a manner identical to the approach used in a previous study [9]. Analysis was performed using Psychophysiological Analysis Software 3.3 (Computational Neuroscience Laboratory of the Salk Institute, La Jolla, CIA, USA) and in-house programs implemented in MATLAB 2021 (MathWorks, Natick, MA, USA). EMG data were sampled during the pre-stimulus baseline and stimulus presentation periods in each trial. A blinded coder checked the video data and confirmed that the participants did not generate large motion artifacts. For each trial, the data were rectified, baseline-corrected with respect to the mean value over the pre-stimulus period, and averaged using intervals of 1000 ms. The data of all film conditions were concatenated, then standardized within each individual. For the EDA and finger temperature data, the analysis method was identical to that for the EMG data, except that the data were not rectified.

*Statistical analysis.* Data analysis was performed using JASP 0.14.1 [46] and MATLAB 2021 software (MathWorks). To test the individual-level linear associations between subjective continuous ratings and physiological activity, Pearson’s product-moment correlation coefficients (*r*-values) were calculated between the ratings and physiological signals for each participant. The associations of valence–corrugator EMG, valence–zygomatic EMG, valence–trapezius EMG, arousal–EDA, arousal–fingertip temperature, and arousal–trapezius EMG were analyzed based on a priori interest as described in the Introduction. The *r*-values were normalized using Fisher transformation, and then analyzed using one-sample and paired *t*-tests (two-tailed), as in previous studies (e.g., [47]). Such two-stage random-effects analyses can demonstrate generalizability across individuals [34]. The results were considered statistically significant at *p* < 0.05. In addition, Bayesian one-sample and paired *t*-tests [48] were conducted to interpret null findings. Results were considered substantial at a Bayes factor (BF) > 3.0 or <0.3 [49]. To visually illustrate the relationships between subjective emotional ratings and physiological activity at the group level, group mean values and regression lines were depicted for subjective ratings and physiological signals.

To test individual-level non-linear associations between subjective continuous ratings and physiological activity, polynomial regression analysis was performed using first degree (linear), second degree (quadratic), third degree (cubic), and fourth degree (quartic) models, with the ratings as the dependent variable and physiological activity as the independent variable. The optimal model was selected based on the adjusted *R*^2^ and root mean squared error (RMSE), for which larger and smaller values indicated better goodness of fit, respectively. The parameters were estimated for each participant and then assessed evaluated at the group level.

## 3. Results

### 3.1. Overview

Figure 1 shows the mean (±standard error) scores for overall valence and arousal ratings and the average physiological activity during each film (see also Appendix A). Visual inspection of the overall ratings suggested the expected linear and U-shaped differences across films in terms of valence and arousal ratings, respectively. Figure 2 shows the group mean time courses of continuous valence and arousal ratings and physiological activity (see also Appendix A). The figures indicated that the emotional film clips elicited dynamic changes in subjective emotional experiences and physiological activity. Figure 3 shows group mean scatterplots and regression lines for the associations of interest between continuous subjective ratings and physiological activity. The expected associations were seen between valence ratings and facial (corrugator and zygomatic) EMG, and between arousal ratings and EDA and fingertip temperature, whereas the relationship between trapezius EMG activity and overall valence and arousal ratings was less clear.

### 3.2. Correlations between Subjective Ratings and Physiological Activity

The correlation coefficients (*r*-values) between the ratings and physiological activity were calculated for each participant, for analysis of intra-individual subjective–physiological association of interest (Figure 4). After the Fisher *z* transformation, the *r*-values were subjected to one-sample *t*-tests against zero (Table 1). The results replicated previously reported significant associations of valence–corrugator EMG (negative), valence–zygomatic EMG (positive), and arousal–EDA (positive) (*t*(29) > 2.99, *p* < 0.01, *d* > 0.54). In addition, the negative association between arousal ratings and fingertip temperature was significant (*t*(29) = 2.88, *p* = 0.007, *d* = 0.53). The association of valence–trapezius EMG and arousal–trapezius EMG was not significant (*t*(29) < 0.61, *p* > 0.54, *d* < 0.11). Bayesian one-sample *t*-tests showed that these null results were substantial (BF < 0.23), suggesting no association between continuous valence or arousal ratings and trapezius EMG activity.

Next, the strength of associations of trapezius EMG and fingertip temperature with subjective ratings were compared with the strength of that of facial EMG and EDA using paired and Bayesian-paired *t*-tests (Table 2). When the predicted directions were opposite (e.g., arousal–EDA vs. arousal–fingertip temperature), the value of one result for each participant was multiplied by –1 prior to assessment of association strength. The results revealed that the negative association of valence–corrugator EMG was significantly stronger than that of valence–trapezius EMG (*t*(29) = 3.09, *p* = 0.004, *d* = 0.83). The association of arousal–EDA was also significantly stronger than that of arousal–trapezius EMG (*t*(29) = 3.31, *p* = 0.002, *d* = 0.61). There was no significant difference between the valence–zygomatic EMG and valence–trapezius EMG associations, or between the arousal–EDA and arousal–fingertip temperature associations (*t*(29) < 1.66, *p* > 0.11, *d* < 0.55). Among these null findings, Bayesian paired *t*-tests showed that the null difference in associations between arousal–EDA and arousal–fingertip temperature was substantial (BF = 0.24), suggesting a similar association with continuous arousal ratings across EDA and fingertip temperature.

### 3.3. Polynomial Regression Modeling of Subjective–Physiological Concordance

The linear and non-linear subjective–physiological associations were assessed using polynomial regression analysis. Linear, quadratic, cubic, quartic models were generated for each participant, with ratings as the dependent variable and physiological activity as the independent variable, and evaluated at the group level.

First, the optimal models were identified based on the adjusted *R*^2^ and RMSE values (Figure 5). The quartic models outperformed the lower-order models with respect to the valence–corrugator EMG, valence–zygomatic EMG, and valence–trapezius EMG associations, according to both indices. For the arousal–EDA, arousal–fingertip temperature, and arousal–trapezius EMG associations, both indices indicated that the linear models had the best fit.

Next, the parametric estimates of the models (Figure 6) were assessed using one-sample *t*-tests against zero (Table 3). Significant linear associations were found for valence–corrugator EMG (negative) and valence–zygomatic EMG (positive) in the optimal (i.e., quartic) models (*t*(29) > 2.43, *p* < 0.03, *d* > 0.44). Significant linear associations were found for arousal–EDA (positive) and arousal–fingertip temperature (negative) for the optimal (i.e., linear) models (*t*(29) > 2.83, *p* < 0.01, *d* > 0.44). No significant associations were found for valence–trapezius EMG or arousal–trapezius EMG (*t*(29) < 0.47, *p* > 0.64, *d* < 0.09).

## 4. Discussion

The present results revealed that facial (corrugator and zygomatic) EMG and EDA were linearly associated with continuous valence and arousal ratings, respectively, while participants viewed emotional films. These results replicate previous findings (e.g., [9]) and indicate that subjective emotional experience dynamics can be assessed using these physiological measures. Furthermore, our polynomial regression analysis newly revealed that the linear relationships were more consistent across participants than the curvilinear relationships.

More important, the results revealed that fingertip temperature was negatively and linearly associated with the continuous arousal ratings. In addition, the comparison with EDA using a Bayesian approach suggested that EDA and fingertip temperature were similarly associated with continuous arousal ratings. These results are in agreement with previous studies showing that fingertip temperature decreased in response to emotionally arousing stimuli (e.g., fearful films [24]) and increased in response to relaxing stimuli (e.g., a sedative odor [28]). The results also coincide with evidence that peripheral skin temperature primarily reflects sympathetic nervous activity [12], as demonstrated for EDA [11]. However, no previous studies tested associations of continuous subjective valence/arousal ratings with fingertip temperature changes. Our study is the first to show that subjective arousal dynamics can be assessed using fingertip temperature in a manner similar to EDA.

We found no significant linear or non-linear associations of subjective valence or arousal ratings with trapezius EMG activity. Additionally, trapezius EMG activity had a weaker association with valence ratings than did corrugator EMG, and a weaker association with arousal ratings than did EDA. These results appear to be inconsistent with previous findings that trapezius EMG activity increased in response to emotionally negative and arousing stimuli (e.g., stress exposure [19]). However, no previous studies investigated associations of trapezius EMG activity with continuous subjective valence or arousal ratings. Visual inspection of trapezius EMG activity during each film (Figure 1) suggested that activity increased when participants watched angry films, but not when they watched sad films. These data suggest that trapezius EMG activity may be activated only in response to highly negative and arousing stimuli. This idea should be further investigated using different stimuli.

Our findings have practical implications. Because emotions influence many aspects of daily life, including behaviors and decision-making [50], the use of physiological signals to assess emotional dynamics could be useful for obtaining objective, unbiased insight into emotions. Although EDA is the gold standard for evaluating arousal, assessment of EDA may be difficult in some individuals, including those with excessive sweating of the palms [51,52]. In such individuals, fingertip temperature may be an alternative parameter for assessing arousal dynamics. In addition, simultaneous measurement of EDA and fingertip temperature, both of which reliably indicate subjective arousal dynamics but provide unique information (e.g., sweat gland- or blood vessel-driven activity), may increase the reliability of arousal dynamics assessment.

The present study has some limitations. First, as room temperature was consistently maintained within the normal range, it remains unknown whether the association between subjective arousal dynamics and fingertip temperature would be observed in cold or hot environments. A previous study has suggested that the starting finger temperature should be above approximately 32 °C to elicit a reduction of fingertip temperature in response to emotional stimuli [24]. Future research regarding associations of fingertip temperature with subjective arousal ratings under various temperature conditions is needed to further validate fingertip temperature analysis for assessing emotion in naturalistic environments. Second, coupling between affect dial ratings and physiological activity was used to assess subjective–physiological concordance, it is not clear how precisely the affect dial ratings reflect changes in subjective emotional experience. The temporal preciseness and individual differences in continuous ratings remain a subject of debate [53]. Further methodological studies are needed to investigate subjective emotional dynamics and subjective–physiological emotional concordance.

## 5. Conclusions

The present findings showed that continuous valence ratings during emotional film observation were not associated with trapezius EMG activity, but were associated with corrugator and zygomatic EMG. Arousal ratings were similarly associated with EDA and fingertip temperature, but not with trapezius EMG. These data newly suggest that fingertip temperature can be used to assess the dynamics of subjective emotional arousal.

## Figures and Tables

**Figure 1 sensors-22-06553-f001:**
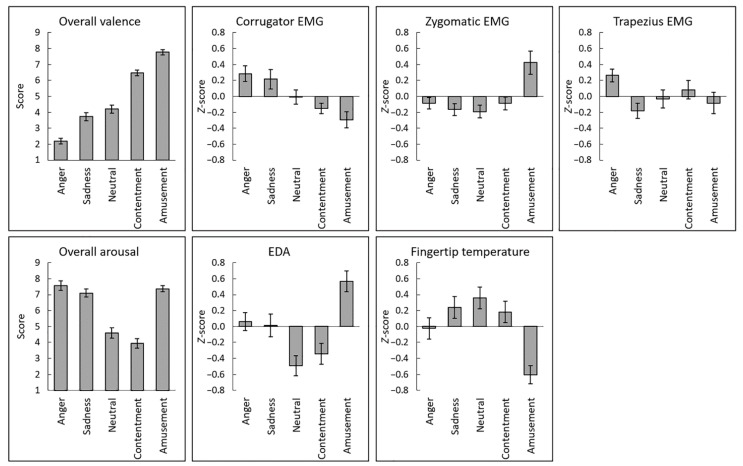
Mean (±standard error) overall ratings of valence and arousal, electromyography (EMG) recorded from the corrugator supercilii, zygomatic major, and trapezius muscles, electrodermal activity (EDA), and fingertip temperature across films. Physiological data were standardized within individuals.

**Figure 2 sensors-22-06553-f002:**
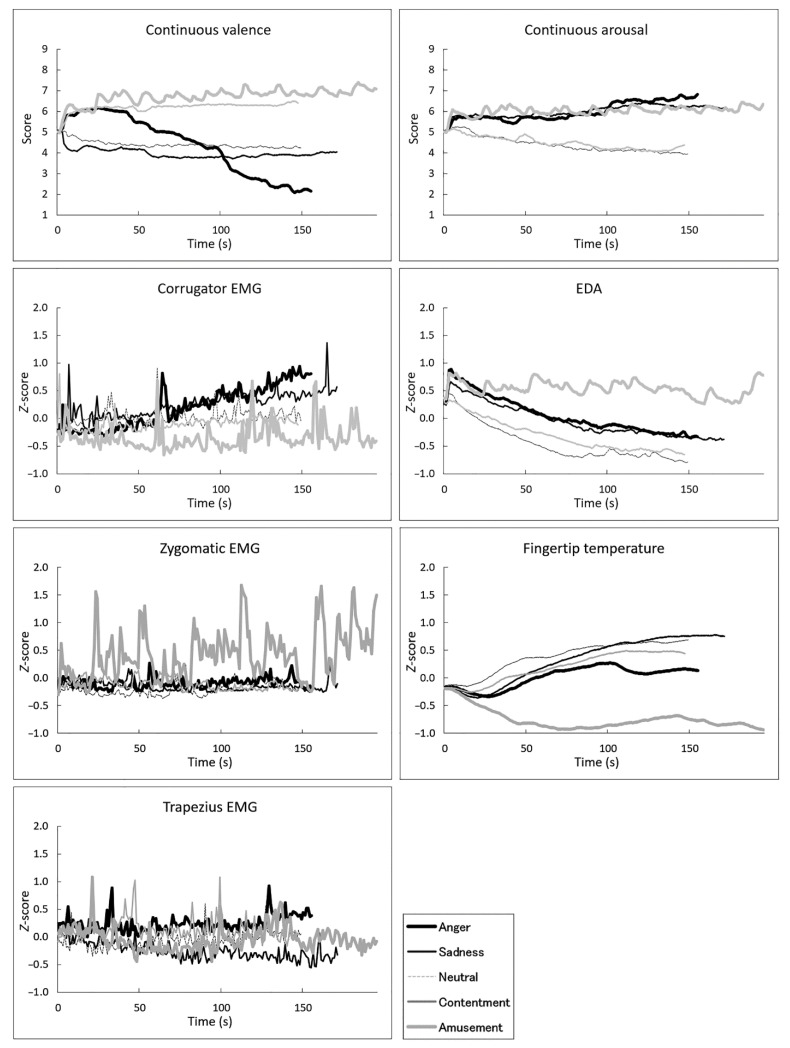
Group-mean continuous subjective ratings of valence and arousal, electromyography (EMG) recorded from the corrugator supercilii, zygomatic major, and trapezius muscles, electrodermal activity (EDA), and fingertip temperature, over time. Physiological data were standardized within individuals.

**Figure 3 sensors-22-06553-f003:**
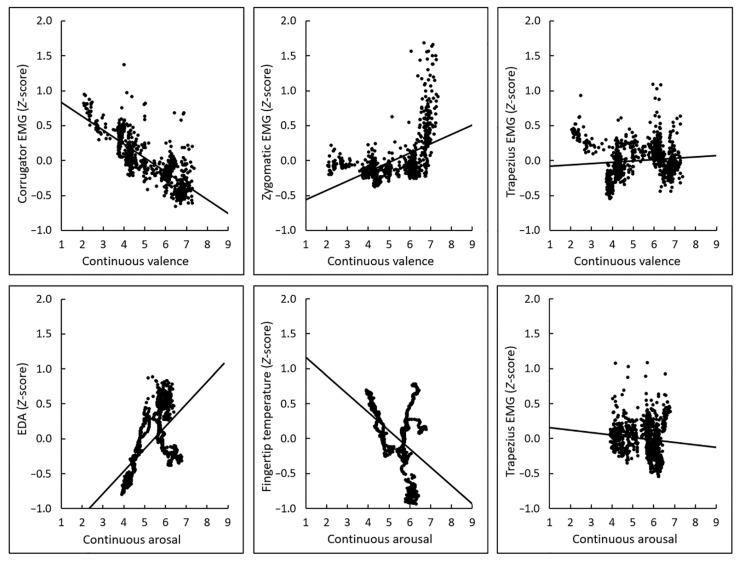
Group-mean scatterplots and regression lines of the expected subjective–physiological concordance over time. Associations were expected between the valence ratings and electromyography (EMG) activity recorded from the corrugator supercilii, zygomatic major, and trapezius muscles, and between the arousal ratings and electrodermal activity (EDA), fingertip temperature, and trapezius EMG. Physiological data were standardized within individuals.

**Figure 4 sensors-22-06553-f004:**
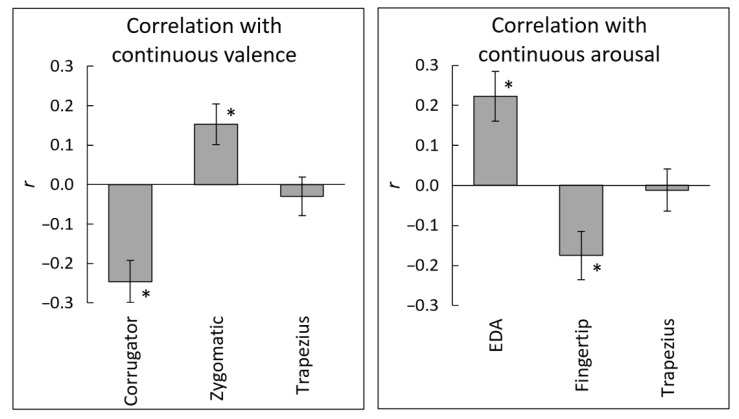
Mean (±standard error) intra-individual correlation coefficients of expected subjective–physiological concordance over time. Associations were expected between the valence ratings and electromyography (EMG) activity of the corrugator supercilii, zygomatic major, and trapezius muscles, and between the arousal ratings and electrodermal activity (EDA), fingertip temperature, and trapezius EMG. ***, *p*
*<* 0.05.

**Figure 5 sensors-22-06553-f005:**
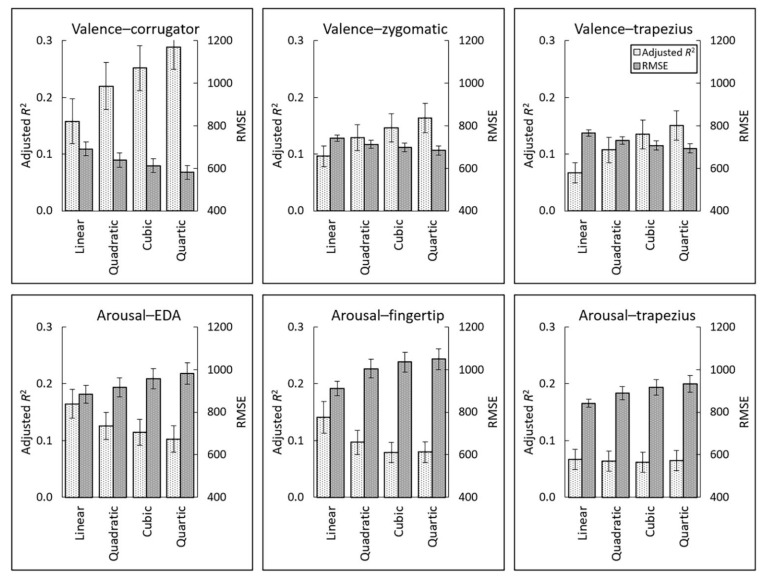
Mean (±standard error) adjusted *R*^2^ and root mean squared error (RMSE) values of linear, quadratic, cubic, and quartic polynomial regression models of expected subjective–physiological concordance. EDA, electrodermal activity.

**Figure 6 sensors-22-06553-f006:**
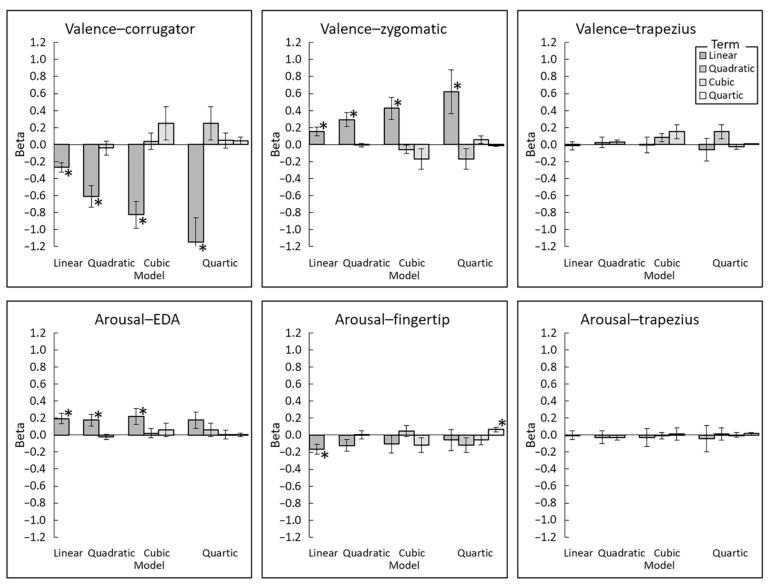
Mean (±standard error) parameter estimates of linear, quadratic, cubic, and quartic polynomial regression models of expected subjective–physiological concordance. *, *p* < 0.05. EDA, electrodermal activity.

**Table 1 sensors-22-06553-t001:** Results of one-sample *t*-test and Bayesian one-sample *t*-tests of subjective–physiological associations.

Association	*t*-Test			Bayesian
	*t*	*p*	*d*	BF_10_
Valence–corrugator	**4.54**	**<0.001**	**0.83**	**281.28**
Valence–zygomatic	**3.00**	**0.006**	**0.55**	**7.45**
Valence–trapezius	0.61	0.549	0.11	**0.23**
Arousal–EDA	**3.61**	**0.001**	**0.66**	**29.21**
Arousal–fingertip	**2.88**	**0.007**	**0.53**	**5.85**
Arousal–trapezius	0.23	0.822	0.04	**0.20**

BF_10_, Bayes factor in favor of an alternate hypothesis over the null hypothesis. EDA, electrodermal activity. All tests were two-tailed. There were 29 degrees of freedom for all one-sample *t*-tests. Significant (*p* < 0.05) and substantial (BF > 3.0 or <0.33) results are in bold.

**Table 2 sensors-22-06553-t002:** Results of paired *t*-test and Bayesian paired *t*-tests of subjective–physiological associations.

Comparison	*t*-Test			Bayesian
	*t*	*p*	*d*	BF_10_
Valence–corrugator vs. valence–trapezius	**3.09**	**0.004**	**0.83**	**9.13**
Valence–zygomatic vs. valence–trapezius	1.65	0.109	0.55	0.66
Arousal–EDA vs. arousal–fingertip	0.67	0.508	0.12	**0.24**
Arousal–EDA vs. arousal–trapezius	**3.31**	**0.002**	**0.61**	**14.91**

BF_10_, Bayes factor favoring the alternate hypothesis over the null hypothesis. EDA, electrodermal activity. All tests were two-tailed. All paired *t*-tests had 29 degrees of freedom. Significant (*p* < 0.05) and substantial (BF > 3.0 or <0.33) results are in bold.

**Table 3 sensors-22-06553-t003:** Results of one-sample *t*-test (two-tailed) for the parameter estimates in polynomial regression modeling of subjective–physiological associations.

Association	Model	Term											
		Linear			Quadratic			Cubic			Quartic		
		*t*	*p*	*d*	*t*	*p*	*d*	*t*	*p*	*d*	*t*	*p*	*d*
Valence–corrugator	Linear	**4.92**	**<0.001**	**0.90**									
	Quadratic	**4.70**	**<0.001**	**0.86**	0.51	0.613	0.09						
	Cubic	**5.25**	**<0.001**	**0.96**	0.41	0.683	0.08	1.11	0.277	0.20			
	Quartic	**4.00**	**<0.001**	**0.73**	1.27	0.215	0.23	0.51	0.615	0.09	1.00	0.327	0.18
Valence–zygomatic	Linear	**2.99**	**0.006**	**0.55**									
	Quadratic	**3.48**	**0.002**	**0.64**	0.32	0.752	0.06						
	Cubic	**3.21**	**0.003**	**0.59**	1.12	0.271	0.21	0.26	0.794	0.05			
	Quartic	**2.43**	**0.022**	**0.44**	1.38	0.178	0.25	1.22	0.231	0.22	1.62	0.117	0.30
Valence–trapezius	Linear	0.26	0.800	0.05									
	Quadratic	0.39	0.698	0.07	1.36	0.185	0.25						
	Cubic	0.05	0.960	0.01	1.76	0.089	0.32	0.34	0.736	0.06			
	Quartic	0.46	0.649	0.08	1.77	0.087	0.32	1.07	0.295	0.20	0.56	0.583	0.10
Arousal–EDA	Linear	**3.12**	**0.004**	0.57									
	Quadratic	**2.48**	**0.019**	0.45	0.77	0.447	0.14						
	Cubic	**2.34**	**0.026**	0.43	0.40	0.692	0.07	0.90	0.375	0.17			
	Quartic	1.78	0.085	0.33	0.78	0.442	0.14	0.13	0.896	0.02	0.06	0.954	0.01
Arousal–fingertip	Linear	**2.83**	**0.008**	**0.52**									
	Quadratic	1.78	0.086	0.33	0.01	0.990	0.00						
	Cubic	0.94	0.354	0.17	0.72	0.475	0.13	0.31	0.758	0.06			
	Quartic	0.47	0.640	0.09	1.32	0.197	0.24	0.91	0.370	0.17	**2.55**	**0.016**	**0.47**
Arousal–trapezius	Linear	0.01	0.989	0.00									
	Quadratic	0.37	0.717	0.07	1.24	0.224	0.23						
	Cubic	0.27	0.789	0.05	0.18	0.859	0.03	0.18	0.859	0.03			
	Quartic	0.27	0.789	0.05	0.19	0.851	0.04	0.01	0.989	0.00	1.38	0.177	0.25

All tests had 29 degrees of freedom. Significant results (*p* < 0.05) are in bold. EDA, electrodermal activity.

## Data Availability

The data supporting the findings of this study are available within the Appendix A.

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
