# Peer review of "Exploration of Emotion Dynamics Sensing Using Trapezius EMG and Fingertip Temperature"

_sensors, 2022, doi:10.3390/s22176553_

Round 1

Reviewer 1 Report (Previous Reviewer 2)

This manuscript (id: sensors-1893504) represents a revised version of the manuscript (id: sensors-1721787) which I reviewed on May 18, 2022.

In the revised manuscript, the authors have adequately addressed the remarks from my previous review report and I believe that it has been sufficiently improved to warrant publication in Sensors. With respect to the previous version, an additional author was engaged in the manuscript, and his contribution (i.e., analysis and writing) is clearly indicated (p. 13, l. 413-414) and in line with the changes in the manuscript.

A typo: “de-pendent”, p. 5, l. 235.

Author Response

Dear Reviewer,

Thank you for constructive suggestion.

Point 1
A typo: “de-pendent”, p. 5, l. 235.
Response
As suggested, we have corrected the typo (l. 235) 

Yours sincerely,

Wataru Sato

Reviewer 2 Report (Previous Reviewer 1)

This paper aims to investigate the correlation between certain physiological signals (trapezius EMG and fingertip temperature) and continuous subjective ratings of induced emotions. The results supported the conclusion that trapezius EMGs were not associated with the dynamics of emotional valence and arousal, while confirmed that fingertip temperature could be used as a good indication for emotion recognition. The manuscript is well written, and the results are well presented.

My previous concerns are addressed through the revision. No further questions to the authors.

Author Response

Dear Reviewer,

Thank you for your patience, and your continued interest in our work.

Yours sincerely,

Wataru Sato

This manuscript is a resubmission of an earlier submission. The following is a list of the peer review reports and author responses from that submission.

Round 1

Reviewer 1 Report

This paper is a follow-up work for Sato et al., 2020. The previous work developed the methods of using physiological signals to assess the dynamics of subjective emotional valence and arousal, and has shown that facial EMGs, EDA and nose-tip temperatures are good sources for such assessment. This paper investigated more candidate signals that are better for practical use with the same methods as the previous work, and got some meaningful findings which could address some ambiguous observations in previous studies. Overall, the manuscript is well written. In the introduction the author clearly stated the rationale of studying trapezius EMG and fingertip temperature by introducing the limitation in previous studies. The experimental design and data processing were appropriate and rigorous. The statistical methods were valid and correctly applied. The results supported the conclusion that trapezius EMGs were not associated with the dynamics of emotional valence and arousal, while confirmed that fingertip temperature could be used as a good indication for emotion recognition.

Below are my comments:

Page 4, Line 170 - 171: It’s not very clear for the description of the length of the recordings. Does it mean you recorded 10-s data during each period (baseline and film stimulus presentation) respectively or 10-s data in total over the two periods? Please consider to rephrase.

Page 5, Line 194 – 198: Need to expand. Currently there’s only one sentence mentioning each figure briefly, which looks more like a caption. Please describe the contents in each figure in more detail and also add some interpretations to the results.

Reviewer 2 Report

The manuscript reports on a study intended to examine the correlation between selected physiological signals and continuous subjective ratings of induced emotions. It describes experimental settings involving 30 subjects (16 women, mean age 22.6 with the standard deviation of 2.7 years) in which emotions were induced by exposing them to video clips, and reports the following results:

- Result 1: Electromyography of the corrugator supercilii and zygomatic major muscles is associated with continuous valence ratings.
- Result 2:  Electrodermal activity (skin conductance level) of the palm is associated with continuous valence ratings.
- Result 3: Fingertip temperature is negatively associated with continuous arousal ratings.
- Result 4: There are no significant associations of subjective valence or arousal ratings with electromyography of the trapezius muscle.

Remarks:

1. The novelty of the study is not apparent:

1a. The associations reported in Results 1 and 2 have already been demonstrated in one of the author’s previous studies conducted in similar experimental settings (which was adequately cited in the manuscript, cf. Reference 5).

1b. The association which is very similar to that reported in Results 3 was demonstrated in other studies (again, these studies were adequately referenced). The author emphasizes the following novelty: the subjective ratings in the observed study are continuous. However, it is not clear why the fact that subjective ratings are continuous is important (please cf. also Remark 2).

1c. Three out of five emotional stimuli (i.e., films) used in the experiment was produced in the scope of a previous study, and it remains unclear how the remaining two stimuli were produced (cf. p. 3, l. 106-113).

2. Although the continuous ratings appear convenient for the statistical analysis applied in this study, it has not been shown to what extent the utilized cued-recall (i.e., a posteriori) assessment based on computer mouse movements sampled at 10Hz (cf. p. 3-4, l. 134-139) was chronologically precise.

3. The statistical analysis in this study (e.g., linear regression,  etc.) is conducted under the (implicit) assumption that there is a linear relationship between the selected physiological signals and continuous subjective ratings. However, this assumption is not supported in the manuscript.

4. The subjects used 9-point scales to assess emotional valence and arousal (cf. p. 3, l. 123-125 and 130-132), and the author considers the group mean continuous subjective ratings. However, it is not clear whether (and if so, how) it was ensured that different subjects use the same values to describe the same emotional states.

5. In addition, Fig. 1 and 2 show the mean overall ratings for very specific emotional states: anger, sadness, neutral, contentment and amusement. How these emotional labels were derived? The reader may assume that each of five video clip stimuli was associated with a particular emotional state (e.g., the authors briefly state that “these stimuli were previously validated in a Japanese sample [32]”, p. 3, l. 107-108). If so, why is it justifiable to assume that a given video clip would induce the same emotional state in different subjects?